# SARS-CoV-2 as an Oncolytic Virus Following Reactivation of the Immune System: A Review

**DOI:** 10.3390/ijms24032326

**Published:** 2023-01-24

**Authors:** Joao P. Bounassar-Filho, Laura Boeckler-Troncoso, Jocelyne Cajigas-Gonzalez, Maria G. Zavala-Cerna

**Affiliations:** 1School of Medicine, Health Sciences Decan, Universidad Autonoma de Guadalajara, Zapopan 45129, Mexico; 2Immunology Research Laboratory, Health Sciences Decan, Universidad Autonoma de Guadalajara, Zapopan 45129, Mexico

**Keywords:** SARS-CoV-2, oncolytic virus, tumor reduction, tumor remission, lymphoma, NK cells, Hodgkin’s, COVID-19

## Abstract

The effects SARS-CoV-2 inflicts on human physiology, especially in patients who developed COVID-19, can range from flu-like symptoms to death, and although many lives have been lost during the pandemic, others have faced the resolution of aggressive neoplasms that once proclaimed a poor prognosis following traditional treatments. The purpose of this review was to analyze several fortunate case reports and their associated biomolecular pathways to further explore new avenues that might provide oncological treatments in the future of medicine. We included papers that discussed cases in which patients affected by COVID-19 suffered beneficial changes in their cancer status. Multiple mechanisms which elicited a reactivation of the host’s immune system included cross-reactivity with viral antigens and downregulation of neoplastic cells. We were able to identify important cases presenting the resolution/remission of different aggressive neoplasms, for which most of the time, standard-of-care treatments offered little to no prospect towards a cure. The intricacy of the defense mechanisms humans have adopted against cancer cells through the millennia are still not well understood, but SARS-CoV-2 has demonstrated that the same ruinous cytokine storm which has taken so many lives can paradoxically be the answer we have been looking for to recalibrate the immunological system to retarget and vanquish malignancies.

## 1. Introduction

Since 12 December 2019, when the first symptoms of a new severe upper respiratory disease were reported, later giving rise to the COVID-19 pandemic, the myriad of effects that SARS-CoV-2 can inflict on human physiology have not only been devastating, but also beneficially unforeseen. The vast collection of reported clinical and molecular manifestations have increased due to SARS-CoV-2 exceptional swift capacity to undergo mutations, thus perpetuating the pandemic through the emergence of new variants [1]. Viruses are known obligate intracellular pathogens that replicate by taking advantage of the cellular DNA and RNA machinery, thus modifying the host’s genomic configuration. This ability has been associated with its potential to become pro-oncogenic; which has been observed for, e.g., the Epstein–Barr virus, the Human Herpesvirus-8, and the Human Papillomavirus, amongst others [2]. However, as a contradictory and fascinating phenomenon, previously diagnosed cancer patients who contracted COVID-19 experienced a reduction and/or remission of certain tumors without administration of any type of chemotherapeutic, radiotherapeutic or surgical interventions [3]. The field of biotechnology has demonstrated advances with the use of genetically modified adenoviruses as vectors to target cancer cells, evoking oncolytic effects and promoting further immunogenic responses that engage the adaptive immune system to efficiently penetrate tumors’ microenvironments, while sparing healthy cells [4]. Neoplastic cells are known to elicit a multitude of mechanisms which allow them to successfully escape both innate and adaptive immunity. Some of the contraptions employed by tumor cells to evade immunity checkpoints include the downmodulation and disabling of antigen presentation, the development of immunologic barriers, the downregulation of tumor suppressor genes (i.e., p53, PTEN), hindrance of the programmed cell death pathway (PD-1/PD-L1), T cell tolerance/co-inhibition/dysfunction leading to immune exhaustion and consequently allowing the unbridled division and growth of neoplasms. Interestingly, SARS-CoV-2 infections have shown in some cases to reactivate the immune system via the cytokine storm-dependent inflammation elicited when the virus is actively replicating. In one instance, such immune reconditioning prompted the reactivation of previously exhausted T lymphocytes to “reset” and target tumor cells for destruction (i.e., reduction/remission of acute myeloid/lymphoblastic leukemias) [5]. Conversely, through the path of downregulation/deactivation, certain neoplastic blood disorders underwent tempering and significant reduction (i.e., NK cell lymphomas) [6]. The full mechanisms by which SARS-CoV-2 may induce the reduction and eventual remission of certain tumors are still unknown. The purpose of this review was to analyze papers and case reports that described the potential beneficial side effects of the infection by SARS-CoV-2 and/or the development of COVID-19, such as diminution of cancer disease burden through associated molecular pathways, and to explore new avenues that might provide novel oncological treatments in the future of medicine.

## 2. Discussion

The studies included in this review consisted of three systematic reviews, one meta-analysis, seven case reports, and two case series. A summary of the cases is presented in Table 1. Evidence of regression/remission of tumor possibly resulting from SARS-CoV-2 infection and COVID-19 disease via different molecular mechanisms was described in nine articles, of which, eight were case reports. The remaining case was related to the anticancer immunity observed after mRNA-1273 COVID-19 vaccination. 

Although the combined diagnosis of a malignant neoplasm along with COVID-19 might seem like an unfortunate coalescence, our analysis of multiple case reports surfaced a silver lining that could lead to novel treatment options for cancer. In principle, an individual’s immune system should be capable of defeating malignancies through various immune-dependent mechanisms that include both innate and adaptive responses. However, unfortunately, in the presence of certain pathological microenvironments, the host’s defense is unable to optimally perform and, rather, is deviated towards the path for tolerance. Downregulated pathways developed by cancer cells due to the accumulation of driver mutations hinder the cytotoxic effects of immune cells from exerting their effector functions. Some of the tolerance mechanisms interfering with tumor clearing include the downregulation of HLA-I, the promotion of transforming growth factor-B (TGF-B) and vascular endothelial growth factor (VEGF) activity, and the release of IL-10 and IL-6 by cancerous cells, leading to the development and proliferation of malignancies [5]. Unfortunately, in many cases when tumors are finally detected, their progression has reached such levels of self-sufficiency that render standard treatment therapies (chemotherapy, radiotherapy, immunotherapy, hormone therapy, among others) unable to fully penetrate cell niches and at times bring about more detrimental effects than overall benefits to the patient. It is under these circumstances that an alternative form of treatment, in which one’s own immune system can respond to tumors, becomes of critical significance. Such alternative forms of treatment can comprise antitumor immune responses followed by an infectious process, leading to an excessive production of proinflammatory cytokines which will in turn activate and enhance antigen presentation by dendritic cells, macrophages, B cells, and NK cells. Thus, potentially reanimating the immune system and allowing it to overcome its exhausted state prompted by the tumor [7]. In this review we will summarize the mechanisms described in cases of observed tumor regression following COVID-19.

### 2.1. NK Cell Activation in Response to SARS-CoV-2

This mechanism was demonstrated in a case report where an extremely rare spontaneous resolution of a pituitary microadenoma occurred secondary to an immune response to COVID-19 [8]. It has been speculated that when patients have an infectious process, such as COVID-19, in the presence of a cancer diagnosis, a scenario is created in which an excessive inflammatory response, generated by the virus, ultimately overwrites the primary tolerant environment of the innate immune system, resulting in spontaneous resolution of the tumor. Evidence of such antitumor response has been previously documented in the setting of infectious processes (pneumonia and *Clostridium difficile* colitis) prior to COVID-19, leading to a spontaneous regression of a diffuse large B-cell lymphoma of the maxillary sinus [13]. Another case report also indicated that SARS-CoV-2 triggered an anti-tumor immune response, inducing the remission of a stage III Hodgkin lymphoma through cross-reactivity of viral-specific T cells with tumor antigens, as well as natural killer cell activation by inflammatory cytokines produced in response to the SARS-CoV-2 infection [9]. This mechanism is represented in Figure 1. A third case report, although not associated with an infection, but rather with the administration of an mRNA vaccine for COVID-19, referred to the same activating mechanism. In this case, the authors demonstrated that, following the second boost of vaccination, a patient with a previous myoepithelial cancer of the left parotid gland and lung metastasis underwent shrinkage of the lung metastasis from 50% to 73% after 9 months, without additional treatment. The tumor immune microenvironment was analyzed by mass cytometry in the pre- and post-vaccination setting, identifying important immunological changes, majorly the infiltration of T cells and NK cells, which in contrast to the pre-vaccination samples, demonstrated infiltration with M2 macrophages and neutrophils [7]; importantly, M2 macrophages would be associated with the tolerant state that favors tumor growth. 

### 2.2. Molecular Mimicry or Cross-Reactivity

Another possible mechanism explaining the regression of tumors was presented in a review article, stating that molecular mimicry could constitute an important role in tumor reduction and remission. SARS-CoV-2 proteins, including the Spike protein, are mimicked by markers displayed on tumor cells’ surface such as heat shock proteins (HSP60, HSP70), therefore eliciting a similar immune response. This mechanism of cross-reactivity could allow cytotoxic T cells and antibodies specifically produced against the virus to also target the cancer cells, leading to the regression of the tumor [14]. However, there is still a missing link, since more than often, the mimicry of antigens is not enough, and the addition of danger signals, which usually represent the result of a real tissular insult, is necessary for the development of a full and robust immune response [15]. In the context of COVID-19, the insult becomes real; therefore, this proposed mechanism following acute infection might be sufficient for DCs to promote T cells activation. With respect to changes in the microenvironment, the development of a bystander activation might be the route that makes cells prone to become more easily activated. The characterization of such danger signals, in the context of tumor-elicited immune responses, needs to be further investigated to develop a real strategy for innovative treatments. 

### 2.3. SARS-CoV-2 Viral Entry through ACE-2/NRP-1 following Destruction by Cytotoxic T Cells

A different mechanism was described in a case series, which presented three patients affected by metastatic colorectal cancer showing evidence of tumor reduction amidst COVID-19 [10]. It is widely accepted that SARS-CoV-2 infects cells through an interaction of the receptor binding domain (RBD) in the spike (S) protein with the human receptor ACE-2 [16]. Although ACE-2 receptors are commonly known to be present in alveolar pneumocytes, they have also been observed in other tissues, allowing the infection of other types of cells, including the ones in the colon [17].

The interaction between SARS-CoV-2 and colon cancer cells was hypothesized to happen trough a previously described mechanism that, most likely, guides SARS-CoV-2 infectivity pleiotropism, consisting of an interaction between the S1 protein with the transmembrane receptor neuropilin-1 (NRP-1). Interestingly, co-expression of ACE2 and NRP1 on a human colon adenocarcinoma cell line (Caco-2) led to an increase in infectivity when compared to that observed with cells that only expressed the ACE2 receptor [18]. Tumor reduction, in these cases, could be explained by the interaction between SARS-CoV-2 and colon cancer cells through the ACE-2/NRP-1 receptors. The interplay between the virus and the colon cancer cells is guided by the presence of the complex ACE2/NRP-1, allowing an effortless infection to take place and ultimately leading to a direct immune response and the destruction of the tumor cells by cytotoxic T cells. In this regard, the presence of ACE2 receptors and NRP-1 in different tissues could be beneficially seen as a prognostic marker in patients with certain cancer types and in favor of the usage of SARS-CoV-2 as an oncolytic agent, which will be discussed in a later section of this review. Figure 2 provides a schematic representation of this mechanism, with more detail. 

### 2.4. Downregulation of NK Cells via the Expression of Inhibitory Molecules after Infection by SARS-CoV-2 through the ACE2 Receptor

Another mechanism in which a deep exploration should be conducted to induce the reduction/remission of certain neoplasms is downregulation. In deranged hematological malignancies, such as NK lymphomas, the neoplastic cells develop evasion mechanisms that will inhibit the host’s immune response from direct targeting by CD8+ T cells. However, neoplastic NK cells display an array of inhibitory molecules that can be targeted for downregulation. In the NK lymphoma remission case report [6], SARS-CoV-2 was able to infect both healthy and neoplastic NK cells via entry through the ACE2 receptor to then induce the expression of the inhibitory receptor NKG2A in great quantities, which led to their exhaustion, a reduction in cytotoxic activity, and even the induction of apoptosis (Figure 3). The abundant number of receptors that NK cells can express on their surface makes them great candidates for a swift upregulation as well as a downregulation. The NKG2 receptor family almost always becomes activated by the HLA-E ligand, and this mechanism can follow two pathways: activation via NKG2C/E/D, especially NKG2D [19], or deactivation via NKG2A, as previously mentioned. The Ly49 membrane receptor family is also involved in the activation or inhibition of NK cells, especially with regards to self-tolerance and antiviral and antitumor states. The KIR family (Killer cell Immunoglobulin-like Receptor) can also activate both pathways of upregulation and downregulation [20]. Due to the rapidness in which NK cells can elicit antitumor immunity, it is important to explore mechanisms involving SARS-CoV-2 as a stimulator of tissue-specific cytokines to induce the activation of NK cells and promote their infiltration of tumor niches (which, sometimes, cannot be reached by cytotoxic drugs) to obliterate cancer cells. 

Another case report of a complete remission of a follicular cell lymphoma amidst COVID-19, illustrated the shrinkage of a para-aortic lymph node [11]. This evidence supports the hypothesis that an important inflammatory process could reanimate the immune system. Although it was seen that an immune response was elicited after SARS-CoV-2 infection, causing the partial destruction of the tumor, there was no clear description of the molecular/immunological mechanisms towards its remission. 

### 2.5. Proposed Mechanisms for SARS-CoV-2 as an Oncolytic Virus

As the field of biotechnology grows, genetically engineered viruses have taken important roles in novel treatment therapies. We propose that a genetically modified version of SARS-CoV-2, in which virulence is significantly diminished, could be used to infect a cancer patient as a form of treatment. This therapeutic mechanism could follow both pathways of upregulation as well as downregulation, depending on the imminent malignancy. In addition, infecting a patient with SARS-CoV-2, which is known to be a lytic virus, will promote the release of its content (including DNA, ribosomes, proteins, and various organelles), thus triggering the immune system via tumor-associated antigens to target the other malignant cells [12]. Although this proposal might sound controversial, it might be helpful for patients with a poor prognosis of terminal neoplastic illnesses, against which conventional methods have little or no effect. In the presence of a cytokine storm, many antigens are released, which can promote cross-reactions between viral proteins and tissue-specific neoplasms, where the virus may achieve successful replication. This would increase antigen presentation by dendritic cells, macrophages, and B cells. NK cells activation can be promoted as well by cytokine production. Such an environment would give an opportunity to exhausted CD8+T cells to be upregulated and reactivated via the expression of certain genes such as LAG-3, responsible for the regulation of exhausted cytotoxic T cells [21]. Among the effector functions of CD8+ T cells, it is important to mention the release of IL-2 (recruiting NK cells) and interferon-gamma (disrupting tumor proliferation and angiogenesis). Unfortunately, it is challenging to predict which cytokines will be produced in which quantities, since it all depends on the uniqueness of one’s immune response, but a common denominator that must be addressed in this treatment proposal is the management of IL-6 as well as the blockage of PD-L1. IL-6 is a proinflammatory cytokine with pleiotropic biological activities, including the induction and maintenance of B cell modulation and Th17 cell differentiation, among other actions related to acute inflammation, such as the induction of acute-phase reactants synthesis by the liver [22]. IL-6 is known to elicit the most dangerous effects of the COVID-19 cytokine storm, as well as to dampen the recruitment and development of lymphocytes; therefore its blockade was suggested as a possible therapy, especially for patients with a severe infection [23]. In addition, IL-6 also promotes the development, progression, and dissemination of tumors via the induction of angiogenesis and of intracellular adhesion molecules. PDL-1 is expressed on the cell surface of a variety of aggressive neoplasia and is a key player in promoting the exhaustion of T cells. 

Therefore, the addition of IL-6 blockers, as well as of a PD-L1 inhibitor could be useful in cancer treatment whenever previous testing for PD-L1 expression on the resident cells of a tumor has been demonstrated, so that the inhibitor will truly have an effect [24]. This is of utmost importance, as PDL-1 inhibitors can elicit dangerous autoimmune side effects. The combination of treatments could also decrease the appearance of side effects previously attributed to these biologics by reducing the required dosage of PDL-1 and IL-6 inhibitors. We believe that a “controlled” infection with a modified version of SARS-CoV-2 along with the administration of IL-6 blockers and PD-L1 checkpoint inhibitors can promote the regression of several different difficult-to-treat aggressive malignancies. 

## 3. Conclusions

Several mechanisms have been identified in the context of tumor regression amidst COVID-19. At least nine case reports/series were identified in which patients experienced such desirable clinical outcome, which makes the proposal of this article worth considering for the design of future strategies that can successfully treat malignancies which harbor difficulties related to the tumor type or location. Furthermore, a case report with detailed tissue examination confirmed tumor remission after mRNA vaccination for COVID-19. Further characterization of the triggers for a robust immune response is required to design a “perfect fit” for each case. The use of novel biological treatments that can modulate the immune response such as IL-6 inhibitors and immune point blockers could aid in this strategy of a personalized alternative for the treatment of malignancies, with less side effects and higher efficacy.

## Figures and Tables

**Figure 1 ijms-24-02326-f001:**
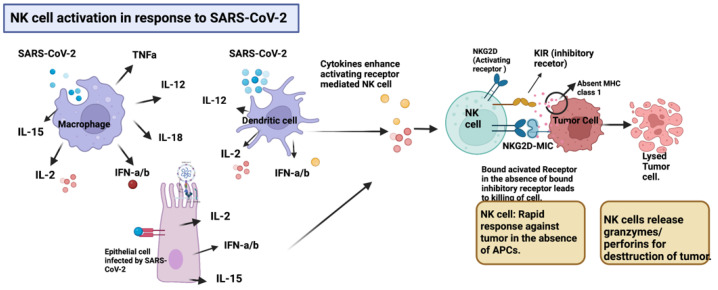
Inflammatory cytokines comprising IL-12, IL-15, IL-2, IFN-α, and INF-β produced by various cells including macrophages, dendritic, and epithelial lung cells in response to SARS-CoV-2 upregulate the expression of activating receptors (NKG2D) in NK cells, promoting a rapid response against tumor cells in the absence of APCs. The downregulation of MHC class I presentation by tumor cells as an evasion mechanism, triggers the activation of NK cells. Upon activation, NK cells release granzymes and perforins, which destroy the tumor. Created with BioRender.com.

**Figure 2 ijms-24-02326-f002:**
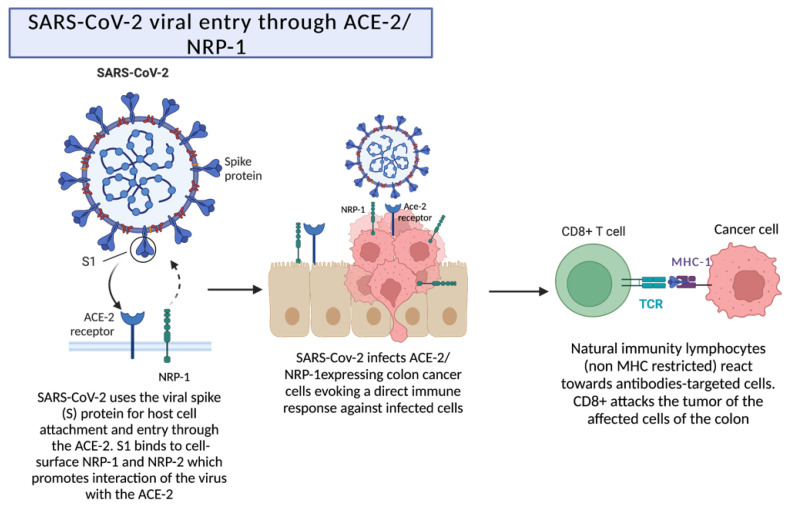
Reduction of a tumor by the interaction between SARS-CoV-2 and ACE-2/NRP-1 receptors. ACE-2 receptors are present in alveolar pneumocytes and in other tissues, which allows the entry and attachment of the virus. The presence of the coreceptor neuropilin-1 promotes the interaction of the virus with the ACE-2 receptor, which leads to a direct immune response. Created with BioRender.com.

**Figure 3 ijms-24-02326-f003:**
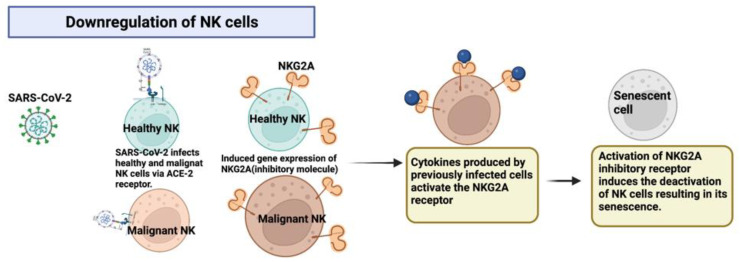
SARS-CoV-2 infects both healthy and malignant NK cells and promotes the expression of the inhibitory molecule NKG2A, thus causing cell senescence. Created with BioRender.com.

**Table 1 ijms-24-02326-t001:** Summary of Case Reports and Case Series of patients with remission after SARS-CoV-2 exposure.

Reference	Diagnosis before COVID-19	COVID-19 Clinical Course	Clinical Evaluation after COVID-19
[5]	57-year-old woman with acute myeloid leukemia M2 with 11q23/KMT2A abnormality	Before initiation of chemotherapy, the patient reported fever, pulmonary symptoms, and progressively worse hypoxemia, with a positive PCR for SARS-CoV-2. Treated in the ICU with remdesivir and dexamethasone IV for 10 days, then treated for COVID-19-related pulmonary fibrosis	Two months after the initial diagnosis, blood cell counts improved, returning to normal. Bone marrow sampling to reevaluate leukemia revealed a normocellular marrow with 55% cellularity and <5% of blasts. Final diagnosis: spontaneous morphologic remission in the absence of disease-modifying therapy for acute leukemia. One month later, the FISH analysis revealed a normal result and proved the molecular cytogenetic remission. Eight months later, the patient had hematological recurrence of the primary disease.
[6]	20-year-old male with relapsed/refractory NK/T cell lymphoma associated with Epstein–Barr virus and autoimmune hemolytic anemia	The patient presented with a 5-day history of fatigue, fever, cough, and dyspnea. O_2_ saturation was 93%. Chest tomography showed diffuse bilateral ground-glass opacities, and lab studies indicated thrombocytopenia, leukocytosis, severe anemia, elevated CRP, and a positive test for COVID-19. Treated with methylprednisolone and O_2_	Eleven days after the onset of COVID-19, the patient presented with a spontaneous steady clinical improvement, with hemolytic markers and platelets count normalization. Reduction in the number of both healthy and malignant NK clonal cells and increase in CD8+ T cells. Dropped values in plasma EVB-DNA, from 229,876 copies/mL to 495 copies/mL. Final diagnosis: remission of NK lymphoma during the COVID-19 infection. Two months after the COVID-19 infection, a recurrence was observed.
[7]	61-year-old woman with T2NOMX metastatic myoepithelial carcinoma of the left parotid gland.	Six months after the diagnosis, the patient received the 2nd booster of the mRNA-1273 COVID vaccine, with severe side effects (fever, chills, fatigue, myalgias, muscular weakness, headache, and mental fogginess for 7 days).	Evidence of persistent tumor shrinkage on CT scans: 50%, 67%, and 73% reduction at 3, 6, and 9 months, respectively, after the second dose of the vaccine.
[8]	32-year-old man with a pituitary microadenoma and secondary adrenal insufficiency and scotoma, managed with steroid therapy for 2 years	Febrile with shortness of breath, pulmonary rhonchi, and crepitation, increase in blurry vision and scotoma. O_2_ saturation was 80%. Patient’s white blood cell count, CRP levels, D-dimer, lactate dehydrogenase increased. Treated with methylprednisolone. After 10 days, the patient presented improvement but remained febrile for 1.5 months	Three months after the resolution of the COVID-19 infection, the patient underwent a control MRI which showed improvement in the pituitary microadenoma. The changes included disappearance of the hypointense lesion and hyperintense enhancement that were seen in the previous MRI (6 months before). Clinically, the patients’ blurry vision improved, as well as the headaches.
[9]	61-year-old male with stage III Hodgkin lymphoma caused by Epstein–Barr virus	SARS-CoV-2 pneumonia with a course of 11 days of supportive ward-based care without any treatments	Remission of Hodgkin lymphoma in four months, with widespread resolution of lymphadenopathy and reduction of metabolic uptake. PCR showed EBV viral reduction to 413 copies/mL (log^10^ 2⋅62)
[10]	Patient 1, a 65-year-old man with a pT4apN1b metastatic colorectal adenocarcinoma. Patient 2, a 58-year-old man with a pT3pN0 RAS-mutated, metastatic colorectal adenocarcinoma Patient 3, a 60-year-old woman with a pT3pN2A metastatic colorectal adenocarcinoma	Patient 1 developed severe COVID-19 symptoms and Patients 2 and 3 developed mild symptomatic COVID-19	Patients 1 and 2: CT scan results showed regression of metastatic liver lesions secondary to the immune response induced by COVID-19. Patient 3 experienced a reduction in peritoneal and lung disease by CT scan after recovery from COVID-19.
[11]	61-year-old patient with follicular lymphoma treated with R-bentdamustine	Patient with COVID-19 bilateral pneumonia and partial response to R-bendamustine.	After COVID-19 recovery, a complete remission was observed with a CT-guided biopsy performed twice and by a second follow-up scan.
[12]	Patient 1, a 63-year-old female with acute myeloid leukemia Patient 2, a 28-year-old previously treated for a diagnosis of T acute lymphocytic leukemia	Patient 1 presented with fever, dyspnea, and wheezy chest and was diagnosed with moderate COVID-19 by PCR. Her treatment for AML was postponed until she recovered from COVID-19; she received azithromycin and prednisone for five days. Patient 2 presented with fever, headache, malaise, sore throat, dry cough, loss of smell and taste, and multiple cervical lymphadenopathies. A nasopharyngeal swab confirmed COVID-19 infection. The patient was treated with azithromycin and prednisone for five days.	Patient 1 showed an improvement after 5 weeks of recovery from COVID-19. Her CBC showed a hemoglobin level of 8.7 g/dL, TLC was 1.6 × 100,000, with no blast cells. BMA: blast count was reduced to 3% with explicit dysplasia in the trilineage and normal karyotype. After 3 rounds of repeated investigations, the final diagnosis was a reduction of blast cells after COVID-19, with no relapse until last visited. Patient 2 showed improvement after 6 weeks of recovering from COVID-19. The patient’s cervical lymphadenopathy disappeared, and the total lymphocyte count decreased from 28 × 10^3^/UL and 30% of blast cells to 6.5 × 10^3^/UL, with no evidence of atypical cells or relapses.

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
