# Peer review of "SARS-CoV-2 as an Oncolytic Virus Following Reactivation of the Immune System: A Review"

_ijms, 2023, doi:10.3390/ijms24032326_

Round 1

Reviewer 1 Report

In this review, Joao and other authors summarized the papers that discussed cases in which patients affected by COVID-19 suffered beneficial changes in their cancer status and discussed the multiple mechanisms underlying this effect.

The main context is well organized, and the new form of treatment for cancer using a genetically modified version of SARS-CoV-2 they proposed is novel. There are only two issues associated with publication.

1.    From line 40-43, “The possibility of virus to change DNA’s genomic configuration has been associated with its potential to become pro-oncogenic such as Epstein-Barr virus, the Human Herpesvirus-8, and the Human Papillomavirus, amongst others.”, the reference paper is missing for the statement.

2.    In line 153-156, you demonstrated another mechanism of tumor reduction was interaction between SARS-CoV-2 and the colon cancer cells through the ACE-2/NRP-1 receptor. You should give more evidence to support the claim that ACE2 receptors present in different normal and cancer tissues.

Author Response

We thank the reviewer for contributing to the improvement of our manuscript. Here are specific answers/comments to the reviewers' recommendations.

  1. A reference from line 40-43 was added. Sorry for the omission.
  2. In line 153-156, new references and details were added. 

    Thank you for highlighting this, we believe is a very important piece of information for future readers of this topic.

Reviewer 2 Report

Interesting paper that reviews case studies on COVID-induced cancer remission.

I applaud the authors use of schematics for additional explanatory power.

The English requires extensive editing by a native speaker. Many parts were difficult to read and were awkwardly phrased. The paper would greatly benefit from proofreading and editing to make it more concise and readable.

I would suggest the authors include data/figures/pictures from some of the case studies instead of relying totally on schematics.

Under the funding section the authors state that the received no external funding. Does this mean outside of their institution? Typically government/institutional grants are required for lab operations and should be credited, even if they were not dedicated to this particular paper.

Author Response

We appreciate the work of reviewers and thank them for their contribute to improve our manuscript. Here are specific comments to reviewers' observations.

  1. Two of the co-authors are native English speakers. The entire paper was read proof and we make changes accordingly, which can be identified in the newly submitted version of the manuscript.
  2. The case studies presented in this paper were drawn from the literature and we don´t have data or pictures from the actual patients, just from the papers. We made a table to summarize all cases presented in here, we believe this might help to narrow down information from cases retrieved in here.

    3. We agree that other forms of contribution from our institution can also be considered and should be declared, therefore, changes were made accordingly to this section. Thank you for the clarification.